



# Beach-face slope dataset for Australia

Kilian Vos[1], Wen Deng[1], Mitchell D. Harley[1], Ian L. Turner[1], Kristen D. Splinter[1]

[1]Water Research Laboratory, School of Civil and Environmental Engineering, UNSW Sydney, 110 King Street, Manly Vale, NSW 2093, Australia

*Correspondence to*: Kilian Vos (k.vos@unsw.edu.au)

**Abstract.** Sandy beaches are unique environments composed of unconsolidated sediments that are constantly reshaped by the action of waves, tides, currents, and winds. The most seaward region of the dry beach, referred to as the beach face, is the primary interface between land and ocean and is of fundamental importance to coastal processes, including the dissipation and reflection of wave energy at the coast, and the exchange of sediment between the land and sea. The slope of the beach-face is

a critical parameter in coastal geomorphology and coastal engineering, necessary to calculate the total elevation and excursion of wave run-up at the shoreline. However, datasets of the beach-face slope remain unavailable along most of the world's coastlines. This study presents a new dataset of beach-face slopes for the Australian coastline derived from a novel remote sensing technique. The dataset covers 13,200 km of sandy coast and provides an estimate of the beach-face slope at every 100 m alongshore, accompanied by an easy to apply measure of the confidence of each slope estimate. The dataset offers a unique

view of large-scale spatial variability in beach-face slope and addresses the growing need for this information to predict coastal hazards around Australia. The beach-face slope dataset and relevant metadata are available at https://doi.org/10.5281/zenodo.5606217 (Vos et al., 2021).

## 1 Introduction

The world's coastlines are unique geological environments at the interface between land and sea. Along this coastal fringe,

which is often densely populated (Small et al., 2011), we find beaches composed of unconsolidated sediments (e.g., gravel, sand, mud) that are constantly reshaped by the action of waves, currents, winds and tides (Dean and Dalrymple, 2004). A typical beach cross-section, or beach profile, is illustrated in Figure 1. The beach face is the most seaward region of the subaerial beach, which extends from the berm to the low tide water line and is constantly interacting with the uprush and downrush of individual waves and tidal cycles. The steepness of the beach face (tanβ), or beach-face slope, is a key parameter

in coastal geomorphology and coastal engineering due to its control on important coastal processes. Crucially, the beach-face slope controls the elevation of wave run-up and total swash excursion at the shoreline (Gomes da Silva et al., 2020; Stockdon et al., 2006), processes that are of primary importance for the assessment of coastal erosion and inundation hazards along the coastal boundary (Senechal et al., 2011; Stockdon et al., 2007). The beach-face slope parameter is also a useful proxy for surf-zone hydrodynamics in the absence of costly surf-zone bathymetric surveys and can provide insights into beach swimmer

safety (Short et al., 1993) and wave set-up across the surf zone (Stephens et al., 2011).



Despite the importance of the beach-face slope parameter in numerous empirical formulations in coastal engineering (e.g., wave run-up prediction), large-scale datasets of the beach-face slope remain unavailable along most of the world's coastlines. A complementary global dataset of nearshore slopes, defined from Mean Sea Level (MSL) to the 'closure depth' where negligible morphological change theoretically occurs, was recently compiled by Athanasiou et al. (2019). Referring to Figure

1, this slope indicates the cross-shore gradient of the subaqueous (below MSL) profile and is generally much lower in gradient than the beach-face slope.

The steepness of the beach face is closely related to grain size (Bujan et al., 2019), with coarser/finer sediment typically adopting a steeper/flatter beach face, however it is also linked to the morphodynamic beach state, with lower-gradient slopes usually found along high-energy dissipative beaches and steeper slopes along low-energy reflective beaches (Wright and

Short, 1984). The beach-face slope can be measured by using survey techniques such as topographic RTK-GPS measurements, but these methods require human intervention and remain impractical over large spatial scales (regional to continental). In recent decades, Airborne LiDAR technology has significantly increased the spatial coverage of coastal topographic data, from individual beaches to hundreds of kilometres of coastline (e.g., Middleton et al., 2013; Stockdon et al., 2002). However, in the swash zone, these active remote sensing techniques are hampered by the constant alternating of wet and dry phases as water

levels fluctuate at the shoreline under the action of waves and tides slopes (Middleton et al., 2013), limiting the ability to extract beach-face slopes. UAV surveys and aerial photogrammetry are also subject to the same caveat as structure-from-motion techniques fail in the swash zone due to the non-stationary ground target (Pucino et al., 2021; Turner et al., 2016). More recently, novel methods to extract intertidal zone information using publicly available optical imagery and tide models have been developed (Bishop-Taylor et al., 2019; Tseng et al., 2017), considerably increasing our ability to map coastal

topography over large spatial scales. In particular, Vos et al. (2020) introduced a method to specifically estimate the beach-face slope combining instantaneous satellite-derived shorelines with predicted tides and capable of accurately estimating the long-term average slope between MSL and Mean High Water Springs (MHWS), the region highlighted in Figure 1, across a wide range of coastal environments. This new method paves the way for the generation of large-scale beach-face slope datasets to complement the present nearshore slope global dataset and better resolve the coastal topography.

In Australia, a national dataset of coastal topography and bathymetry was recently identified by the marine research, industry and stakeholder communities as the highest ranked priority for wind-waves research (Greenslade et al., 2020). It is noted in this report that a coastal elevation dataset will significantly improve *"our ability to model other hazards such as tsunamis and storm surges, as well as providing information to improve marine management including marine planning, monitoring, research, and emergency response"*. In a similar study Power et al. (2021) also identified data collection from remotely

sensed products (referred to as item DAT3) as one of the key research priorities in coastal geosciences. More specifically, O'Grady et al. (2019) investigated the contribution of wind waves to total water levels along the Australian coastline and concluded that a key limitation for an accurate Operational Coastal Inundation Forecasting System is the lack of a continental-scale beach-face slope dataset. Historically, due to there being no alternative, studies that have focused on continental to



regional scales have adopted uniform beach-face slope values. For example, recent studies investigating the contribution of wave processes to relative sea-level-rise have either used slope-independent runup formulations (Vitousek et al., 2017) or opted for a constant and therefore arbitrary beach-face slope of 0.1 for all of the world's coastlines (Melet et al., 2018) which has attracted criticism of this approach (Aucan et al., 2019).

In this contribution, we use a recently published and validated beach slope estimation technique (Vos et al., 2020) to fill this gap for the Australian continent. We present a new dataset of beach-face slopes spaced at 100 m increments alongshore for

every sandy beach in Australia totalling 13,200 km of sandy coastline. Each beach-face slope estimate is based on the average slope over the past 20 years and is associated with an easy-to-apply measure of confidence. The methodology to estimate beach-face slopes and confidence bands is presented in the next section. A synopsis then follows of the large-scale spatial variability in beach-face slopes around the Australian continent,  and the integration of this dataset with the National Sediment Compartment Framework for Australia (Thom et al., 2018). The paper concludes with a brief section illustrating potential use

cases of this new dataset for coastal management and inundation forecasting studies.

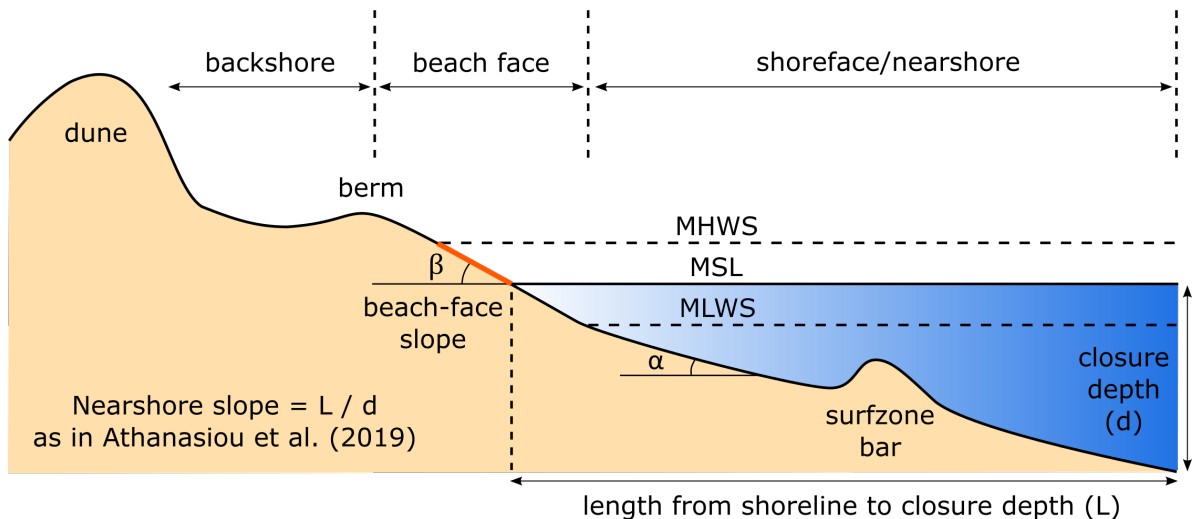

**Figure 1.** Schematic of a beach profile from the dune to the depth of closure, adapted from Coastal Engineering Research Center (1984). The beach-face slope (tanβ) that is mapped in this work is a proxy for the slope of the portion of the profile that is highlighted in orange, extending from Mean Sea Level (MSL) up to Mean High Water Spring (MHWS). The beach-face slope complements the global dataset of nearshore slopes presented in Athanasiou et al. (2019)  that represents the slope extending from the depth of closure up to MSL.

## 2 Methods

### 2.1 Transects dataset

A dataset of shore-normal transects along the Australian sandy coastline was created semi-automatically. Sandy beaches were first extracted from the OpenStreetMaps database (OSM, 2017) and manually quality-controlled in a GIS environment (QGIS

Development Team, 2021). In total, 5,207 sandy beaches were identified along open coasts and semi-enclosed regions. The





sandy shoreline locations were then used to generate 100 m alongshore-spaced cross-shore transects with the "*v_transects*" functionality of GRASS GIS (GRASS Development Team, 2020), generating a total of 132,000 individual transects extending along 13,200 km of sandy coastline circumnavigating the Australian continent. The beach-face slope was estimated along each transect as described in the following section.

**2.2 Beach slope estimation algorithm**

A novel remote sensing technique to estimate beach-face slopes from satellite imagery and modelled tides was applied to each of the 132,000 sandy beach transects. This method is described in detail in Vos et al. (2020) and combines 20-year time-series of shoreline change derived from Landsat imagery with tidal predictions at the time of image acquisition to estimate the slope of the beach face. Briefly, the concept behind this method is that instantaneous shoreline time-series, mapped on images that

were acquired at different stages of the tide, contain a tidal signal that is modulated by the beach-face slope and can be isolated in the frequency domain due to its periodicity. Thus, this technique uses the frequency domain and iteratively seeks a value of the slope that minimises the tidal energy when used for tidal correction. Tidal correction consists of the projection of individual instantaneous shorelines, acquired at different stages of the tide, to a standard reference elevation such as MSL. A simple tidal correction is then applied by translating horizontally the shoreline points along a cross-shore transect using a linear slope:

$$\Delta x_{corrected} = \Delta x + \frac{z_{tide}}{tan\beta} \qquad (1)$$

where $\Delta x_{corrected}$ is the tidally-corrected cross-shore position, $\Delta x$ is the instantaneous cross-shore position, $z_{tide}$ is the corresponding tide level and tanβ is the beach-face slope.

The first step is to map instantaneous shorelines spanning a full 20 years of all available Landsat imagery (Landsat 5, 7 and

8) at every sandy beach and then determine the intersection of individual shorelines with each of the 100 m spaced transects to obtain time-series of (non-tidally corrected) shoreline change, an example of this is shown in Figure 2a for Cable Beach, Western Australia. The time-series were obtained with the open-source CoastSat toolbox — publicly available at https://github.com/kvos/CoastSat and described in Vos et al., (2019). Next, tide levels associated with every shoreline observation are extracted from the global tide model FES2014 (Carrere et al., 2016), as shown in Figure 2b. The tide level

time-series are then transformed to frequency domain, and since these are not evenly sampled (as shorelines are not mapped on all the images because of cloud cover, false detections and other issues), a variant of the Fourier transform, the Lomb-Scargle transform (VanderPlas, 2018) is employed to compute the power spectrum density (PSD). The frequency with the highest peak is then isolated from the PSD, which corresponds to the frequency where the tidal signal (e.g. the spring-neap tidal cycle) is the strongest in the sub-sampled time-series, referred to as the 'peak tidal frequency'. The final step to obtain

the time-averaged estimated beach-face slope at every 100 m alongshore location consists of tidally-correcting the time-series





of uncorrected shoreline change using an iterative range of slope values from 0.01 to 0.2 (from Bujan et al., (2019)), transforming each tidally-corrected time-series to frequency domain and integrating each PSD inside the peak tidal frequency band to construct a curve of tidal energy vs slope (Figure 2c). The best estimate of the beach slope is then the value that minimises this tidal energy in the peak tidal frequency band (Figure 2d).

This beach-face slope estimation technique was validated against *in situ* (beach survey) data along 8 diverse sandy/gravel beaches spanning a broad range of beach-face slopes, tidal regimes, and wave climates (Vos et al., 2020). The validation sites — namely Narrabeen-Collaroy, Moruya-Pedro and Cable Beach in Australia, Duck and Torrey Pines in the USA, Slapton Sands UK, Tairua Beach NZ, and Ensenada MEX — range from microtidal wave-dominated to macrotidal tide-modified beaches, with the *in situ* measured average beach-face slopes varying from $\tan\beta = 0.025$ to $\tan\beta = 0.14$. A strong correlation

($R^2$ of 0.93) was found between *in situ* and satellite-derived beach-face slope estimates, with no systematic under- or over-estimation observed, demonstrating the applicability of this method across a wide range of coastal environments.



**Figure 2.** Flowchart of the methodology to estimate beach-face slopes from satellite-derived shorelines and predicted tides described in Vos et al. (2020). Firstly, instantaneous shorelines are mapped on Landsat imagery with the CoastSat toolbox. The time-series of non-tidally corrected shoreline change (a) and their associated tide levels (b) are then combined in frequency domain to find the slope that, when used for tidal correction, minimises the tidal signal. (c) shows the power spectrum density (PSD) of the tidally-corrected time-series, demonstrating how the slope value modulates the energy inside the tidal frequency band, plotted as a function of slope in (d).


### 2.3 Beach-face slope confidence bands

The tidal energy vs beach-face slope curve shown in Figure 2d is used to quantify the uncertainty of each beach-face slope estimate. When the minimum is very well defined in the curve (as in Figure 2d), this indicates high confidence in the estimate as there is a single value of the slope that clearly minimises the amount of energy in the peak tidal frequency band. However, at other locations, the minimum of the curve is not so well defined, and several beach-face slope values correspond to similar levels of PSD energy, as shown in Figure 3 (from Narrabeen-Collaroy beach located in Sydney, NSW). To incorporate a simple-to-apply measure of uncertainty in the dataset of beach-face slope estimates, a 5% vertical band above the minimum

amount of energy in the curve is used to estimate confidence bands around the slope estimate. To illustrate, Figure 3 shows the 5% band above the minimum energy, the slope that minimises the energy (0.075), and the lower bound (0.065) and upper bound (0.095) slopes that define the 5% of the minimum PSD energy. Note that given the shape of the energy vs beach-face slope curves, which generally tends to start as a parabola and then flatten with an inflection point after the minimum, the confidence bands are not symmetric.

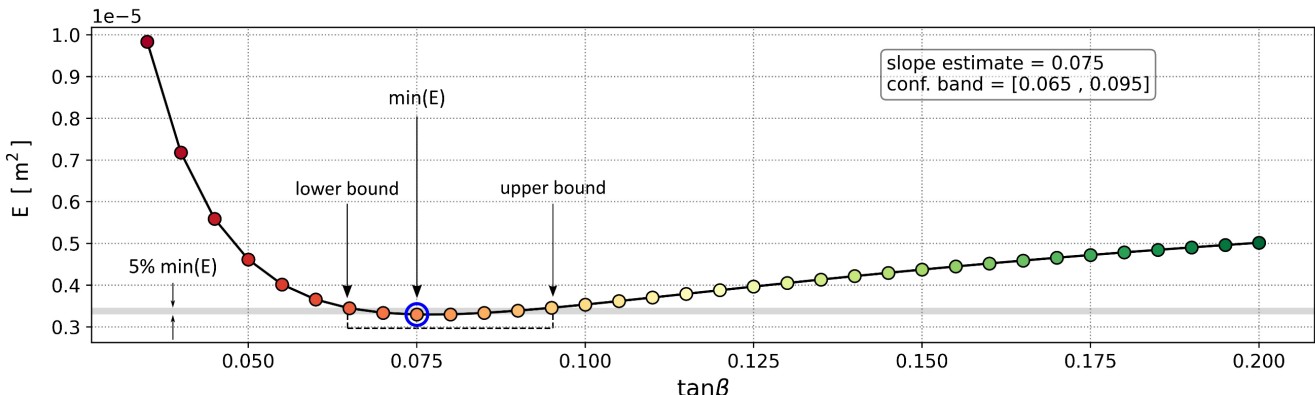


**Figure 3.** Estimation of confidence bands around each beach-face slope estimate based on the tidal energy vs slope curve. The lower and upper bounds are the slopes that are associated with a tidal energy within 5% of the minimum.

### 3 Results

### 3.1 Synopsis of the distribution of beach-face slopes around Australia

Beach-face slopes were estimated at each of the 132,000 beach transects, corresponding to 13,200 km of sandy coastline around Australia. The beach slope data was summarised from the transect scale to the individual beach scale by calculating the weighted average of all the transects at each beach, weighted by the width of the confidence bands to emphasise slopes with higher confidence. The resulting average beach-face slope for every sandy beach around the continent is presented in Figure 4a. Beach-face slope values range between 0.01 and 0.18. The distribution of beach-face slopes for each of the 7 coastal

states in Australia is shown in Figure 4b, indicating median values between 0.055 (Victoria) and 0.08 (Northern Territory), with all 7 interquartile ranges sitting between 0.045 and 0.11.

Along the wave-dominated and energetic southern half of Australia (refer Supplementary Figure 1b), the dataset shows very low gradient beaches (0.01-0.035) along the western coast of Tasmania, western Victoria, and south-west WA. Large spatial variability in beach-face slopes is observed along the tide-modified/tide-dominated northern half of Australia, with

Queensland showing the widest interquartile range (from 0.05 to 0.11).

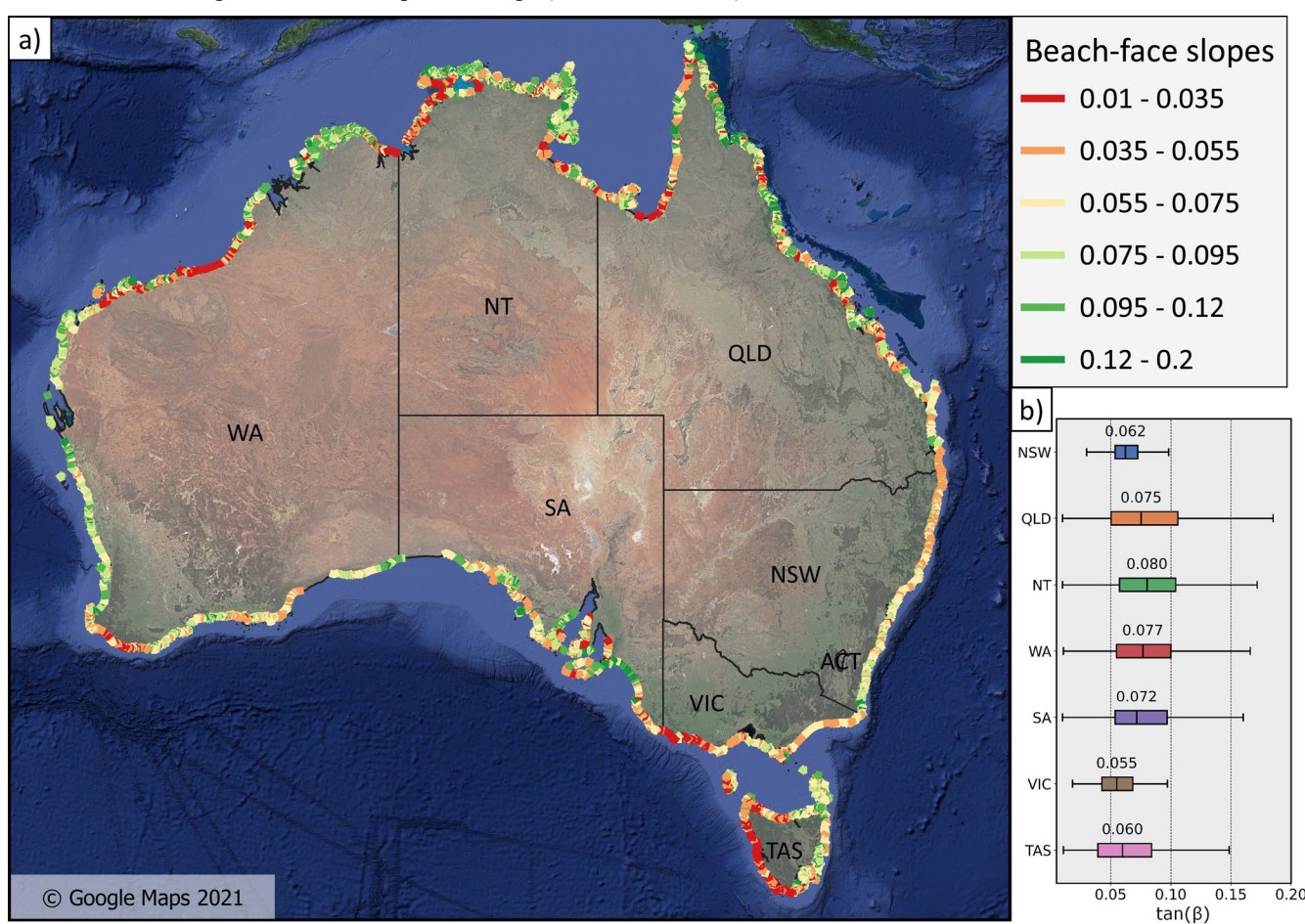

**Figure 4.** Beach-face slopes around Australia along 132,000 100m-spaced transects (equivalent to 13,200 km of coast). (b) shows the distribution for each Australian state.

While State administrative boundaries provide a first division of the country, useful for high-level management purposes, they

are not representative of the geology, surface landforms and shoreline orientation of the Australian coastline. To address this, Thom et al. (2018) proposed a national sediment compartment framework that provides a hierarchical division of the coast integrating inshore/offshore geological factors, major structural landforms such as headlands and changes in shoreline orientation. On this basis, the Australian continent is divided into 23 coastal 'regions', 100 'primary sediment compartments'





and 361 'secondary sediment compartments'. The beach-face slope distribution (weighted-average per beach) for each of the 23 coastal regions is presented in Figure 5. Along the south-eastern coast, narrow slope distributions are observed in Central East (2), Southern NSW (1) and Gippsland (20). Moving northwards, very wide distributions are observed in Central Queensland (3), Eastern (4) and Western Cape York Peninsula (5), and Southern Gulf of Carpentaria (6). The Northern Territory (NT) shows a clear contrast between the relatively steep beaches of East and North Arnhem Land (7-8) and the extremely low gradient Western NT (9). A similar distinction is observed in the northern part of Western Australia (WA), where the manifold of islands in the Kimberley region (10) show steep slopes that contrast with the long and low gradient beaches of the Pilbara region (11). As the tide-modified/tide-dominated coast transitions to wave-dominated, a gradient of decreasing beach slopes is observed along the western regions of Central West WA (12), Southwest WA (13) and Southern WA (14). Wide beach-face slope distributions are observed in South Australia (SA), regions 15-18, which comprises long microtidal coasts, mesotidal gulfs and large offshore islands. The remaining region of the continental mainland is Central and Western Victoria (19), which contains relatively low gradient beaches (median of 0.05). Finally, in Tasmania a clear distinction between the intermediate slopes of the North (21) and East coasts (22) and the West (23) coast is apparent, noting that West Tasmania exhibits the lowest beach-face slopes (median 0.03) of all the coastal regions around Australia.

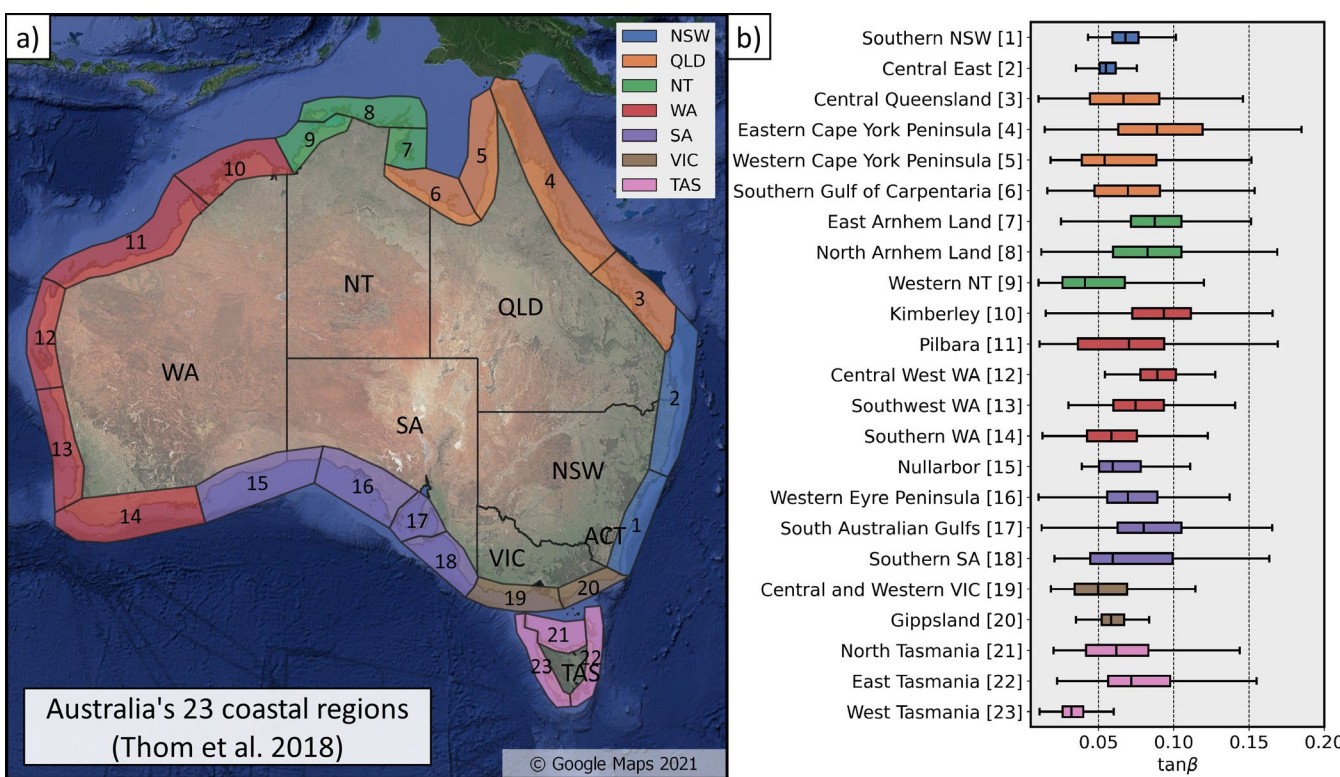

**Figure 5.** Distribution of beach-face slopes for each of the 23 coastal regions identified by Thom et al. (2018).

The widths of the corresponding beach-face slope confidence bands, described in Section 2.3 provide a simple-to-apply metric and filter on the uncertainty in slope estimates. Figure 6a presents a map of the beach-averaged width of the confidence bands, color-coded with a simple three-category 'traffic-light' system: *high confidence* (green) when below 0.025, *medium confidence* (yellow) between 0.025 and 0.05 and *low confidence* (red) above 0.05. Using these categories, 56% of the continental-scale dataset indicate *high confidence*, with 23% *medium confidence* and 21% *low confidence*. Figure 6b shows the distribution of

this confidence metric across each of the 23 regions. Extensive areas of low confidence are observed along the west and south coasts of WA and most of SA, while the remainder of the coastline generally indicates high confidence, interspersed by isolated occurrences of low/medium confidence (e.g., east Arnhem Land, south-east Tasmania). The median width of confidence bands is below 0.025 (i.e., high confidence) for 13 of the 23 coastal regions, between 0.025 and 0.5 (medium confidence) for 5 regions (East Arnhem Land, Central West WA, Western Eyre Peninsula, East/West Tasmania) and above

0.5 (low confidence) for 5 regions including south-west and southern WA and the majority of SA.

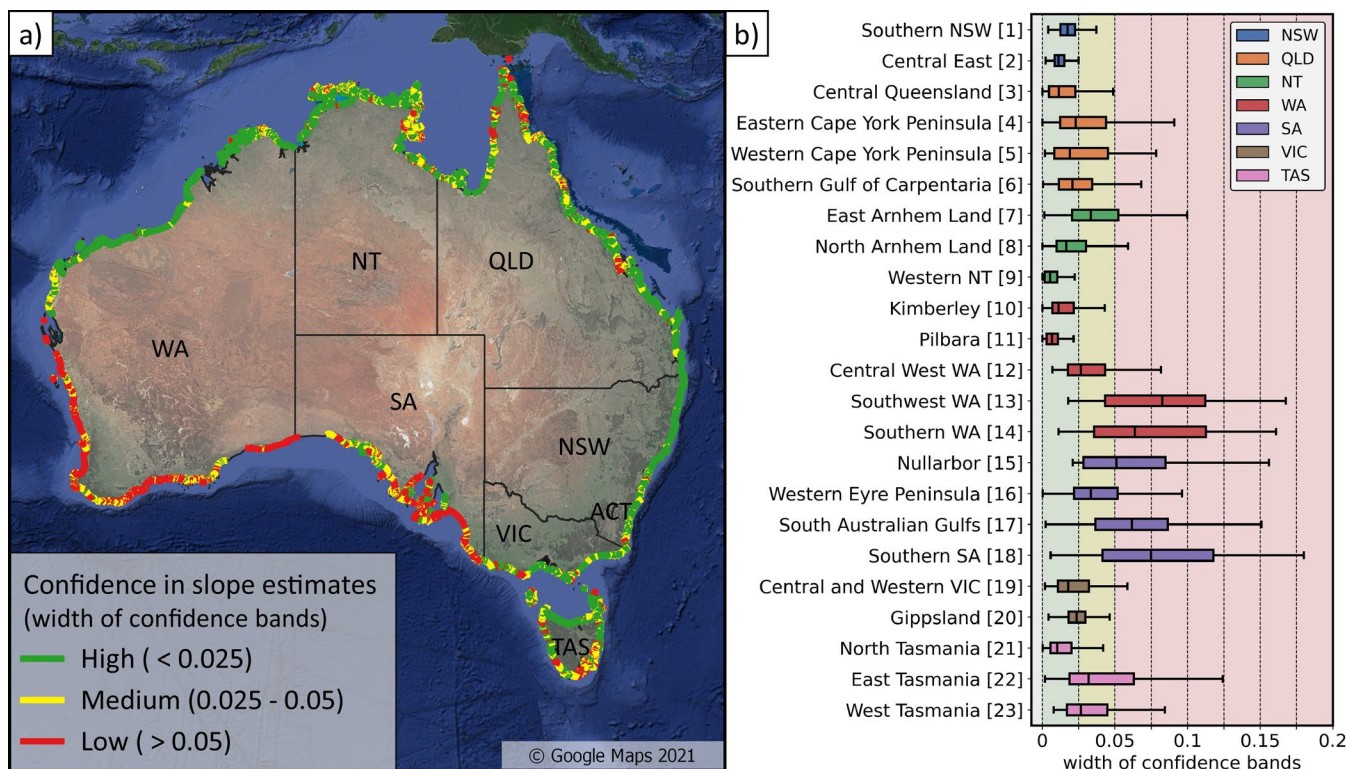

**Figure 6.** Traffic-light system indicating the uncertainty in the beach-face slope estimates. (b) shows the distribution of the width of confidence bands across the 23 regions, indicating on average high confidence across 13 regions, medium confidence across 5 regions (7,12,16,22,23) and low confidence across 5 regions (13,14,15,17,18).

Vos et al. (2020) previously identified that the performance of the beach slope estimation method depends on the signal-to-noise ratio between the accuracy of the satellite-derived shorelines (10-15m based on the validation in Vos et al. (2019b)) and

the horizontal extent of the tidal excursion. Thus, the observation of greater uncertainty in slope estimates along south-west and south WA regions is consistent with the very small tidal range along this coast (< 1 m Mean Spring Tidal Range, see Supplementary Figure 1a). Another issue affecting the signal-to-noise ratio is the aliasing of the tidal signal by sun-synchronous sensors (i.e., Landsat orbits), which means that the full tidal range may not always be captured by the satellite imagery (Bishop-Taylor et al., 2019; Eleveld et al., 2014). Figure 7 shows the percentage of the Mean Spring Tidal Range (MSTR) that is observed by the satellite-derived shorelines at each beach. This analysis reveals that the SA coastal regions (15-18), which exhibit lower confidence in Figure 6, have the lowest tide range coverage with only about 60-65% of the MSTR sampled (Figure 7a). Thus, along the SA regions which are already microtidal (except inside the gulfs), this aliasing results in only part of the tidal range being observed in the available satellite imagery, further reducing the signal-to-noise ratio and leading to lower confidence slope estimates.

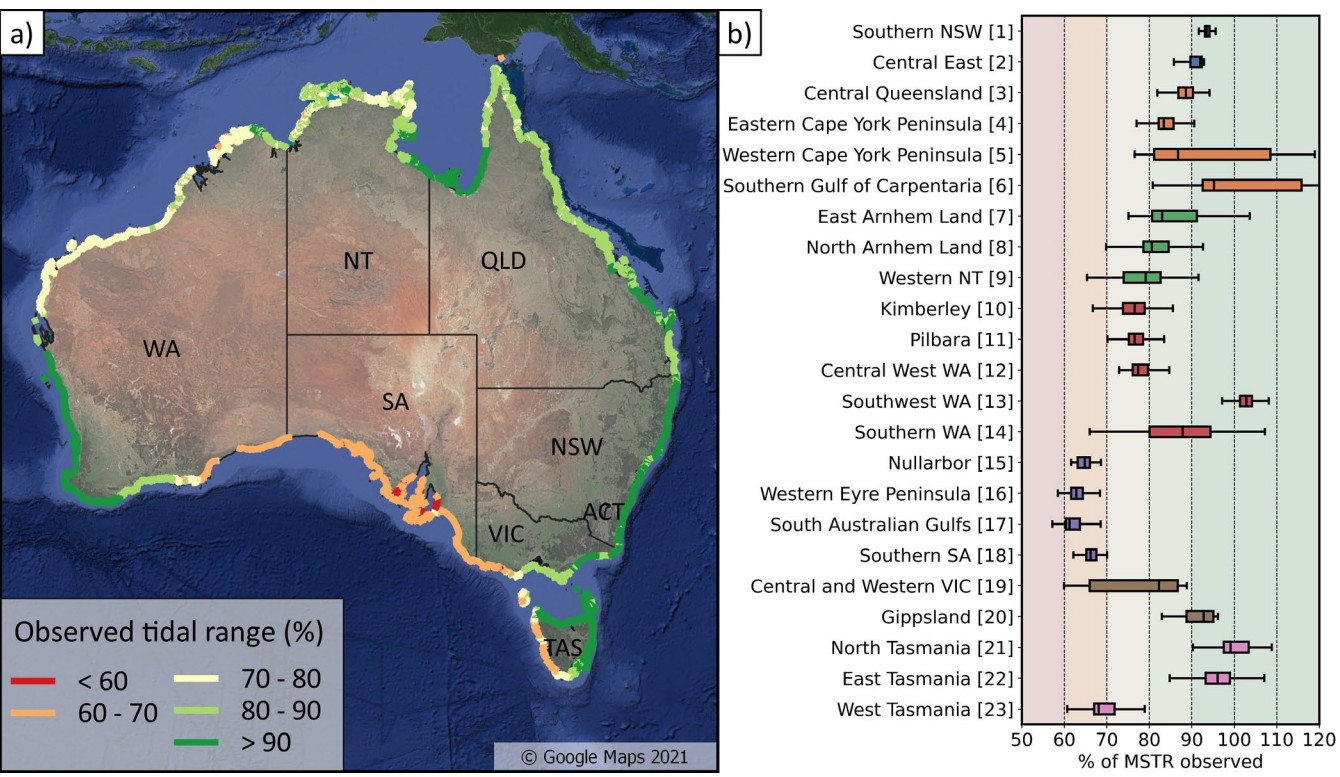

**Figure 7.** Observed tidal range by the satellite-derived shorelines. (a) maps the % of the MSTR that is observed at each beach and (b) shows the distribution of that metric across the 23 coastal regions.

### 3.2 An example embayment-scale application

While the previous section focused on the large-scale distribution of beach-face slopes at the continental scale, this dataset can also provide insights into variations in beach-face slope along any individual embayment or beach. To illustrate, Figure

8a shows beach-face slopes at the transect-scale (100 m spacing alongshore) along a single embayment, the Stockton Bight, located in NSW about 150 km north of Sydney. Stockton Bight is the longest beach in NSW, featuring 32-km of south-facing

sandy coast backed by a large transgressive dune system (Short, 2020). The beach-face slope data at this site, shown in Figure 8a, indicates a distinct alongshore gradient, with lower gradient slopes towards the northern end of the embayment ($\tan\beta$ = 0.04) and steeper slopes in the southern ($\tan\beta$ = 0.1). Interestingly, a sediment grain size dataset at this same site is reported by Pucino (2015), with the median swash zone sand size ($D_{50}$) at 20 equally spaced sample locations (Figure 8b) indicating a distinct gradient in grain size along the Stockton Bight embayment. The observed correlation between grain size and beach-

face slope that is apparent in Figure 8 is in agreement with our understanding of this relationship, generally described by a power-law (Bujan et al., 2019). This example demonstrates that this continental-wide dataset can also be utilised to gain insights into the alongshore variability of beach-face slopes (and, potentially, grain size distribution) along individual beaches and embayments, and inform present-day coastal management and planning.

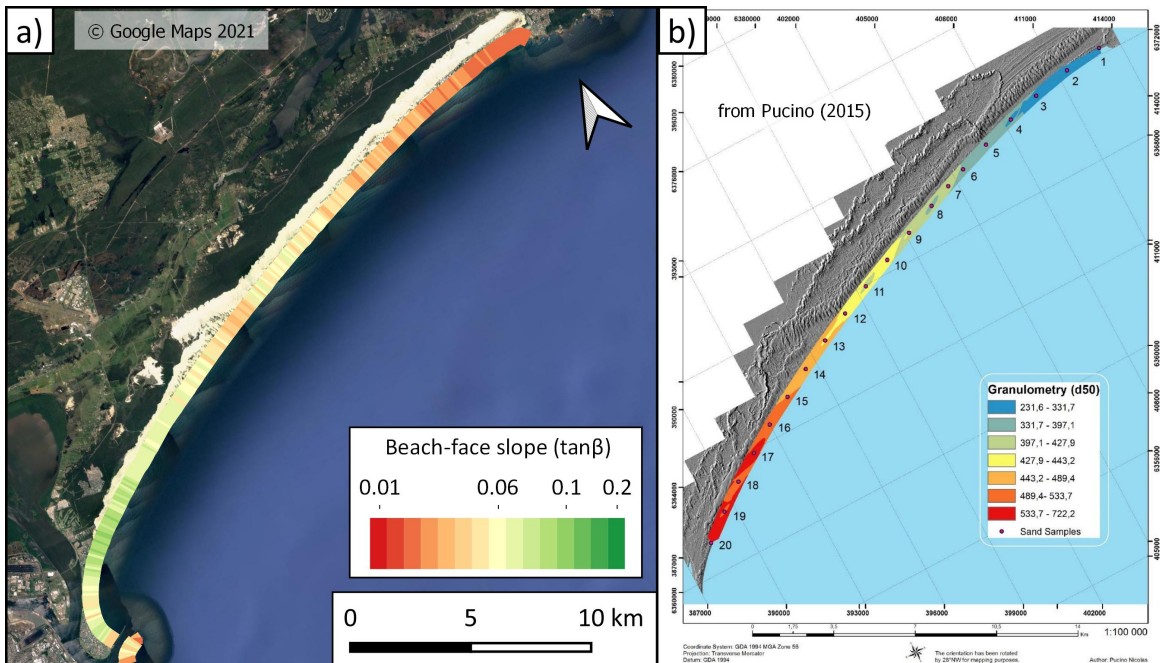

**Figure 8.** Embayment-scale gradient in beach-face slopes and sediment grain size. a) Beach-face slope estimates along 100-m spaced cross-shore transects along the Stockton Bight embayment. b) Figure taken from Pucino (2015) displaying the spatial distribution of median grain size ($D_{50}$) for 20 sand samples along the embayment. $D_{50}$ values are in microns and have been interpolated using inverse-weighted-distance and discretised into 7 quantiles.



### 3.3 Dataset description

The data presented above is made available as two geospatial layers (GeoJSON files): one providing the beach slope estimates on a transect basis (100 m alongshore spacing) and the second providing the estimates averaged over each individual beach. Tables 1 and 2 describe the attributes of each layer, respectively.

At the transect scale (Table 1), the beach-face slope and confidence bands are accompanied by relevant metadata such as the number of shoreline points used to estimate the beach slope, the sediment compartment/region in which the transect is located

and a database id for the transect. A confidence flag (high/medium/low) is associated with each estimate as is illustrated in Figure 6.

At the beach scale (Table 2), the alongshore-averaged slope weighted by the width of the confidence intervals is provided for each individual beach, associated with a confidence flag (high/medium/low) based on the alongshore-averaged width of the confidence bands. Relevant metadata at the beach scale include the Mean Spring Tide Range (MSTR), Median Significant

Wave Height (Hsig), Percentage of Observed Tide Range and beach length.

To assist users of Australia's national sediment compartment framework (Figure 5), the distribution of beach slopes in each sediment compartment is provided in Supplementary Information at both the primary (Supplementary Figure 3) and secondary (Supplementary Figure 4) levels. Geospatial layers containing the primary and secondary compartments are also included in the dataset and displayed in Supplementary Figure 2.


**Table 1. Description of data fields for the beach-face slope dataset at the transect scale**

| Attributes | Values | Description |
|---|---|---|
| *Transect id* | e.g., aus0001-0000, aus0001-0001, … | Database id for each transect |
| *Beach id* | e.g., aus0001, aus0002, …, aus5255 | Database id for each beach |
| *Beach-face slope ($tan\beta$)* | value between 0.01 and 0.2 | Estimate of the beach-face slope (between MSL and MHWS) |
| *Lower confidence bound* | value between 0.01 and 0.2 | Lower limit of the confidence band |
| *Upper confidence bound* | value between 0.01 and 0.2 | Upper limit of the confidence band |
| *Width of confidence band* | value between 0 and 0.19 | Width of confidence band |
| *Number of shoreline points* | value between 100 and 1300 | Number of datapoints in the time-series used for beach-face slope estimation. |
| *Confidence flag* | High/Medium/Low confidence | Quality flag indicating the confidence in the slope estimate at this transect |



| | e.g., Southern NSW, Central Queensland, … | Database id corresponding to the 23 coastal regions as identified by Thom et al. (2018) |
|---|---|---|
| *Coastal region* | e.g., Southern NSW, Central Queensland, … | Database id corresponding to the 23 coastal regions as identified by Thom et al. (2018) |
| *Primary Sediment Compartment* | e.g., NSW01.01, NSW01.02, QLD01.01, … | Database id corresponding to the 100 primary sediment compartments as identified by Thom et al. (2018) |
| *Secondary Sediment Compartment* | e.g., NSW01.01.01, NSW01.01.02, QLD01.01.01, QLD01.01.02, … | Database id corresponding to the 361 secondary sediment Compartments as identified by Thom et al. (2018) |

**Table 2. Description of data fields for the beach-face slope dataset at the individual beach scale**

| Attributes | Values | Description |
|---|---|---|
| *Beach id* | e.g., aus0001, aus0002, …, aus5255 | Database id for each beach |
| *Average beach-face slope ($\overline{tan\beta}$)* | value between 0.01 and 0.2 | Average of the beach-face slope along each transect, weighted by the width of the confidence bands |
| *Average width of confidence bands* | value between 0 and 0.19 | Average width of confidence band over the comprised transects |
| *Average number of shoreline points* | value between 100 and 1300 | Average number of datapoints in the time-series over the comprised transects |
| *Confidence flag* | High/Medium/Low confidence | Quality flag indicating the confidence in the slope estimate at this beach |
| *Coastal region* | e.g., Southern NSW, Central Queensland, … | Database id corresponding to the 23 coastal regions as identified by Thom et al. (2018) |
| *Primary Sediment Compartment* | e.g., NSW01.01, NSW01.02, QLD01.01, … | Database id corresponding to the 100 primary sediment compartments as identified by Thom et al. (2018) |
| *Secondary Sediment Compartment* | e.g., NSW01.01.01, NSW01.01.02, QLD01.01.01, QLD01.01.02, … | Database id corresponding to the 361 secondary sediment Compartments as identified by Thom et al. (2018) |





| | | |
|---|---|---|
| *Mean Spring Tide Range (MSTR)* | value between 0.75 m and 10 m | Mean Spring Tide Range at the beach calculated from the closest grid point in the FES2014 global tide model (Carrere et al., 2016) |
| *Median Significant Wave Height (Hsig)* | value between 0.1 m and 3 m | Median Significant Wave Height from the closest grid point in the CAWCR re-analysis dataset |
| *Percentage of MSTR observed* | value between 57% and 130% | Percentage of the MSTR observed by the satellite-derived shorelines as: $$100\,\frac{\max(tide_{observed}) - \min(tide_{observed})}{MSTR}$$ |
| *Beach length* | value between 75 m and 50 km | Length of each beach or embayment. Very long beaches (> 50 km) were split to optimise memory as images are cropped around each beach |

**5 Conclusion and Outlook**

This study presents a new dataset of beach-face slopes for the Australian coastline derived from a newly-available remote sensing technique. The dataset covers a total of 13,200 km of sandy coast and provides an estimate of the beach-face slope from Mean Sea Level (MSL) to Mean High Water Spring (MHWS) spaced every 100 m alongshore. The beach-face slope estimates are obtained from satellite observations spanning the past two decades and represent the long-term average beach-face slope, and therefore do not capture potential temporal variability. The slope estimation method is based on the value of

the slope that minimises the tidal energy when used to tidally-correct satellite-derived shoreline time-series and has been validated (Vos et al., 2020) across a diverse range of coastal environments. Each slope estimate is accompanied by confidence bands that indicate the upper and lower slope limits that would account for a modulation of the tidal energy within 5% of the minimum. While these are not 95% confidence intervals in the statistical sense, they provide practical information for end-users about the uncertainty associated with any individual slope estimate. Based on the width of the confidence bands. 56%

of the continental-scale dataset are classified *high confidence*, with 23% indicating *mediun confidence* and 21% *low confidence*. Regions of relatively low confidence are predominantly located along the south and south-western coasts. Vos et al. (2020) demonstrated that the accuracy of this method is linked to the signal-to-noise ratio of the satellite-derived shorelines relative to the tidal excursion signal (dependent on the tidal range and the slope itself). Here we show that these low-confidence areas are characterised by the smallest tidal range in the country (MSTR < 1m) in the case of the south-western sector, while

along the South Australia regions only part of the full tidal range (60-70%) is observed by the sun-synchronous satellite sensors as a result of an aliasing effect (Bishop-Taylor et al., 2019; Eleveld et al., 2014).

This new dataset provides critical information for the better understanding of coastal processes around Australia and complements an existing dataset of nearshore slopes measured between MSL and the depth of closure provided by Athanasiou et al. (2019). While the nearshore slopes can be used to transform offshore wave parameters to the nearshore (see Figure 1),

the beach-face slope is necessary to predict the elevation of wave run-up and total swash excursion at the shoreline. For this reason, the beach-face slope dataset is important for improving predictions of coastal erosion and inundation along the Australian coast (O'Grady et al., 2019) and assessing coastal inundation hazards and potential damage to beach-front infrastructure. Furthermore, in the context of an operational coastal flood warning system for sandy beaches (e.g., Leaman et al., 2021; Stokes et al., 2019) the confidence bands associated with each slope estimate are key to propagate the uncertainty

into the run-up equations and provide equivalent confidence bands for the total run-up elevation and swash excursion predictions.

In conclusion, this new dataset offers a unique view of large- to local-scale scale features in beach-face slope variability, providing data in regions with no *in situ* observational coverage. The new dataset can be exploited i) to better understand large-scale geologic factors contributing to the distribution of beach-face slopes and sediment grain size (Bujan et al., 2019;

Short, 2020), ii) to inform coastal management and coastal planning at the embayment scale (e.g., Figure 8), including tourist perception of the beach (Phillips and House, 2009) and beach-front property value (Gopalakrishnan et al., 2011), iii) to improve run-up predictions for an operational coastal inundation forecasting system (O'Grady et al., 2019), iv) to supplement and enhance assessments of beach swimmer safety and surf hazard like the Australian Beach Safety and Management Program (Short et al., 1993).

**Data Availability**

The continental-scale beach-face slope dataset described in this manuscript (Section 3.3) is available in the following Zenodo data repository: https://doi.org/10.5281/zenodo.5606217 (Vos et al., 2021).

**Code Availability**

The source code to map satellite-derived shorelines from Landsat imagery (CoastSat) is available at

https://doi.org/10.5281/zenodo.2779293 (Vos, 2021a). The source code to estimate beach slopes from satellite-derived shorelines and modelled tides (CoastSat.slope) is available at https://doi.org/10.5281/zenodo.3872442 (Vos, 2021b).



**Acknowledgements**

We acknowledge the great efforts by the United States Geological Survey / NASA for providing high-quality open-access data to the scientific community, Google Earth Engine for facilitating the access to the archive of publicly available satellite imagery, and CNES / LEGOS / CLS / AVISO for producing the global tide model FES2014, in particular Frederic Briol for developing the fes Python wrapper. Also thanks to the OpenStreetMap project and contributors (https://www.openstreetmap.org) for their extensive geospatial database. We also thank Nicolas Pucino for collecting and kindly sharing the grain size dataset along Stockton beach. The lead author is supported by a UNSW Scientia PhD scholarship.

**Author Contribution**

K.V., W.D., M.H.D., I.L.T. and K.D.S. devised the study, designed the figures, and wrote the manuscript. W.D. prepared the transect dataset and K.V. processed the beach-face slope data. All authors discussed the results and reviewed the manuscript.

**Competing Interest**

The authors declare that they have no conflict of interest.

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
