# Peer review of "Beach-face slope dataset for Australia"

_Earth System Science Data, 2021_

## Referee Comment (RC1)

**Article Review: *Beach-face slope dataset for Australia**

Floris Calkoen

Preprint of article reviewed here is available at
https://essd.copernicus.org/preprints/essd-2021-388/

**Summary**

The authors present a novel dataset of beach slopes for the coastline of Australia, which they derive by combining satellite-derived shorelines, from Landsat satellite imagery, with tidal levels, from the global tide model FES2014. The method assumes that variability in shoreline positions can be explained two components: 1) a tidal component, driving high-frequency variability; 2) and, a sediment-transport component, driven by processes characterized by lower frequencies. The authors isolate the higher-frequency fortnightly tidal component from the other sediment-transport processes by using the Lomb-Scargle transform. This method to estimate beach-slopes from Landsat satellite imagery was introduced in Vos et al. (2020). The dataset and article currently under review are essentially a a product of the method introduced in that paper.

The following relationship is assumed to relate beach slopes to shorelines positions and tidal heights:

$$\Delta x_{corrected} = \Delta x + \frac{z_{tide}}{tan\beta} \tag{1}$$

, where $\Delta x_{corrected}$ is the tidally-corrected cross-shore position, $\Delta x$ is the instantaneous cross-shore position, $z_{tide}$ is the corresponding tide level and $tan\beta$ is the beach-face slope.

The Lomb-Scargle transformation is used to identify the peak of the fortnightly tidal component in series of tidal levels, sampled according to the timestamp of the satellite images. Following eq. 1, series of shoreline-positions are tidally corrected for a range of beach slope values. For each series in the collection of corrected shoreline-positions the tidal energy is calculated by taking the integral for a set window around the identified peak frequency. Finally, the beach slope is estimated by taking the beach-slope value that minimizes the tidal energy. The dataset was created by applying this method to all OSM-identified sandy beaches, along the Australian coastline, at 100-m alongshore resolution. Uncertainty in beach-slope estimates is expressed by computing the width of 5% upper and lower beach-face slope confidence bounds. The dataset is made available at transect level as well as averaged per beach.

The dataset is an important milestone for making coastal models more data-driven. The authors successfully apply their recently-introduced technique to derive beach-slopes from Landsat satellite imagery. Although their earlier work (Vos et al., 2020) already provided a proof-of-concept

for beach-slope derivation from satellite imagery, this study shows that such method can be applied at continental scale. The article and dataset are understandable and straightforward to use. The dataset is significant (continental scale, high resolution), unique, usuful (science, coastal management, floodrisk modeling and beach safety) and complete (to the best of my knowledge). Nevertheless the manusscript can be substantially improved by addressing some limitations; and, the datasat could be presented more computer-friendly. More importantly, although the temporal frequency of shoreline-position deriviations is essential to the used methodology, the sample data indicate that the repeat cycle is often not 7 or 8 days; this might be an important inconsistency and should be addressed. Overall, I would therefore recommend to return it to the authors for minor revisions.

**1 General comments**

1. In Vos et al. (2020) the authors explain that Landsat data is used because this archive has observations at the same point every eight days. This 8-day revisiting period (or sampling interval) is critical for deriving fortnightly tidal cycles (14.26 days) from the shoreline-position series as in a signal with less frequent observations this tidal cycle would be indistinguishable. Strictly speaking the Landsat repeat cycle is already too long to meet Nequist requirements, and therefore the frequency periodogram shows an aliased pattern.

   Figure 1 shows that at least for Cable beach (toy dataset provided with Vos et al. (2020)) most of the intervals between image acquisition are not eight days. For about 18% the interval is greater than 8 days. When using the algorithm `SDS_slope.reject_outliers()` to remove outliers from the shoreline-position series this figure increases to 39%. Figure 1 shows that the interval between image acquisitions (Cable beach) is typically seven days. It also shows shows that often a revisiting time of zero days is found. In line with these figures, the dataset presently under consideration, has, for some transects 1286 observations between 1999 and 2019—which is 373 more than you would expect from `365.25 (days per year) * 20 (year) / 8 (revisiting period) = 913`.

   Considering the unequally-spaced image-acquisition intervals:

   (a) What causes this highly-unequal temporal distribution of shoreline-position series? This cannot be solely clouds right? In general this data seems to contradict the statement made in Vos et al. (2020) about a target sampling period of no more than 8 days.

   (b) The notebook example provided with Vos et al. (2020) shows that for Cable beach dominant tidal-energy peak is observed at 14.8 days, which is in accordance with the observation that the most frequent image-acquisition interval for this beach is 7 days— enough to observe fortnightly tidal cycles according to the Nequist theorem. However, for many sites the dominant sampling interval might be 8 days (or something else), which results in a peak at 17.5 days (or something else). When creating this dataset, was there a method to automatically determine the peak tidal frequency? If so, how were situations with about an equal amount of 7- and 8 day image acquisition intervals handled? The extraction and use of peak tidal frequency could be addressed in more detail. Presumably some is explained in line 112-114, but it is not entirely clear if this refers to the methods used in Vos et al. (2020) or in this paper.

[Figure]

Figure 1: Interval between image acquisition (days) for time-series with all shoreline positions (left) and time series with outliers removed from the shoreline-position series (right)

(c) Landsat 5, 7 and 8 repeat cycles are all 16 days while Landsat 5 retired when 8 was introduced (*Landsat—Earth Observation Satellites*, 2016), so how can there be so many image-acquisitions with an interval of zero days?

(d) Does the method still work when removing outliers from the series of shoreline positions? If not, this would be useful information to include.

2. The authors state that their method paves the way for the generation of large-scale beach-face slope datasets (line 54-55). Although the beach-face dataset is an important milestone, I think it would be fair to stress a few limitations associated to using Landsat satellite imagery.

(a) The spatially unequal distribution of images in the Landsat archive (Wulder et al., 2016) will hamper derivation of beach slopes in many areas around the globe. For example, many areas in Europe and Africa only have about a quarter of the images as Australia.

(b) Cloud cover has a negative impact on deriving shorelines from satellite imagery, which is also acknowledged by the authors (Line 111). The dataset presents beach slopes for Australia, a continent which is in particular characterized by low mean cloud frequency (Wilson & Jetz, 2016).[1]

Australia can be considered the most favorable continent for beach-slope derivation from Landsat satellite imagery because of its low cloud frequency and large collection of Landsat satellite images. Unfortunately, for many other areas, it will be quite a challenge to derive shorelines from satellite imagery at the same temporal resolution as Australia. As the shoreline-position signal might be too weak to observe forthnightly tidal cycles, I am not sure if it will be feasable to create similar datasets for all other continents around the globe. Maybe it would be fair to note such limitations when discussing the results.

3. Considering the data:
* * *
[1] Viewer available at http://www.earthenv.org/cloud

(a) To ease programmatic access, I recommend to rename identifiers to names without spaces and special characters (e.g., "Average # of shoreline points" is often quite clunky to work with"). I would suggest to use rather short and concise identifiers while providing an extensive description with the metadata. Also try to be consistent in naming[2]: currently some identifiers contain capitals, others do not; most contain spaces, yet others are separated by underscores or hyphens. I would consider just using underscores or PascalCase.

(b) Attribute names in Table 1 do not match identifiers used in the dataset. Make sure the attribute names match the identifiers in the dataset exactly.

(c) In general, when presenting a dataset try to stick as much to the FAIR guiding principles for scientific data management (Wilkinson et al., 2016). For instance, it could be considered to make the data accessible (index-able) according to STAC[3] specification.

**2 Specific comments**

1. Line 84: Not of critical importance, but why not use OSM 2021?

2. Line 112: I believe only the FFT algorithm is not able to deal with unequeally spaced time series, but a regular fourier transformation is able to deal with unequally spaced data. Why not use that one?

**3 Technical corrections**

Overall length of the article is appropriate, although some sections can be written more to the point: paragraph about the relevance of a beach-face slope dataset in the introduction can be written more concisely; and, some sentences the in the conclusion could be moved to the discussion/introduction.

**3.1 Textual**

Following are a few textual suggestions and corrections:

1. Line 37: I would suggest to change "coarser/finer" with "steeper/flaat" to "coarser (finer)" and "steeper (flatter)"; seems to be more in accordance with existing literature.

2. Line 50: "In particular" *and* "specifically" is a bit over the top in this sentence. I would suggest to pick one of these. For example, "Recently, Vos et al. (2020) introduced a method to specifically.."

3. Line 55 - 68: I would rewrite this paragraph more concisely, probably leaving the quoted text out; the importance of a beach-slopes datasat can be stressed in a few sentences.

4. Line 75: "Inundation forecasting" to "flood risk modeling"?
* * *
[2] For example, see https://stackoverflow.com/questions/7662/database-table-and-column-naming-conventions
[3] https://stacindex.org

5. Line 92: "This method is described in detail in Vos et al. (2020) *and* combines.." Conjunction feels a bit weird here. Maybe better to "This method, which is described in detail in Vos et al. (2020), combines .."

6. Line 107-8: Would either leave "open-source" or "publicly" out because they imply each other. Maybe something like "Time-series were obtained with CoastSat, a toolbox publicly available at.."

7. Line 111: I would suggest to also include a reference to the founders of the Lomb-Scargle method (Lomb, 1976; Scargle, 1982) as well as to the comprehensive overview provided by VanderPlas (2018).

8. Line 116: Maybe change "from Bujan et al. (2019)" to "following Bujan et al. (2019)"?

9. Line 205: "which means" to "which implies"?

10. Line 258-262: I would leave the results of Vos et al. (2020) out of the conclusion.

11. Line 268-271: Consider to rewrite this sentence. For example "low-confidence areas, in the south-western sector are characterized by . . . , while..". Further, I would suggest to rephrase the sentence so that references are no longer required—would be more appropriate to include such information in introduction or methodology.

12. Line 255-289: The conclusion contains a lot of information which might be more appropriate in a section discussing the results. Consider to use the conclusion to list the most important findings of this study provide and outlook for the future. For example, the effects of aliasing and the reference to Eleveld et al. (2014), Bishop-Taylor et al. (2019) should be discussed earlier. Similarly, the importance of a beach-slope dataset was already discussed extensively in the introduction.

13. Table 1: "id" should be capitalized.

14. Table 1: In the column "Values", I would leave "confidence" out in "High/Medium/Low confidence".

15. Table 1: Consider to describe the ranges in the "Values" section mathematically instead of with words.

**References**

Lomb, N. R. (1976). Least-squares frequency analysis of unequally spaced data. *Astrophysics and Space Science*, *39*(2), 447–462. https://doi.org/10.1007/BF00648343

Scargle, J. D. (1982). Studies in astronomical time series analysis. II - Statistical aspects of spectral analysis of unevenly spaced data. *The Astrophysical Journal*, *263*, 835. https://doi.org/10.1086/160554

Eleveld, M. A., van der Wal, D., & van Kessel, T. (2014). Estuarine suspended particulate matter concentrations from sun-synchronous satellite remote sensing: Tidal and meteorological effects and biases. *Remote Sensing of Environment*, *143*, 204–215. https://doi.org/10.1016/j.rse.2013.12.019

*Landsat—Earth observation satellites* (USGS Numbered Series No. 2015-3081). (2016). U.S. Geological Survey. Reston, VA. https://doi.org/10.3133/fs20153081

Wilkinson, M. D., Dumontier, M., Aalbersberg, I. J., Appleton, G., Axton, M., Baak, A., Blomberg, N., Boiten, J.-W., da Silva Santos, L. B., Bourne, P. E., Bouwman, J., Brookes, A. J., Clark, T., Crosas, M., Dillo, I., Dumon, O., Edmunds, S., Evelo, C. T., Finkers, R., . . . Mons, B. (2016). The FAIR Guiding Principles for scientific data management and stewardship. *Sci Data*, *3*(1), 160018. https://doi.org/10.1038/sdata.2016.18

Wilson, A. M., & Jetz, W. (2016). Remotely Sensed High-Resolution Global Cloud Dynamics for Predicting Ecosystem and Biodiversity Distributions (M. Loreau, Ed.). *PLOS Biology*, *14*(3), e1002415. https://doi.org/10.1371/journal.pbio.1002415

Wulder, M. A., White, J. C., Loveland, T. R., Woodcock, C. E., Belward, A. S., Cohen, W. B., Fosnight, E. A., Shaw, J., Masek, J. G., & Roy, D. P. (2016). The global Landsat archive: Status, consolidation, and direction. *Remote Sensing of Environment*, *185*, 271–283. https://doi.org/10.1016/j.rse.2015.11.032

VanderPlas, J. T. (2018). Understanding the Lomb–Scargle Periodogram. *The Astrophysical Journal Supplement Series*, *236*(1), 16. https://doi.org/10.3847/1538-4365/aab766

Bishop-Taylor, R., Sagar, S., Lymburner, L., Alam, I., & Sixsmith, J. (2019). Sub-Pixel Waterline Extraction: Characterising Accuracy and Sensitivity to Indices and Spectra. *Remote Sensing*, *11*(24), 2984. https://doi.org/10.3390/rs11242984

Bujan, N., Cox, R., & Masselink, G. (2019). From fine sand to boulders: Examining the relationship between beach-face slope and sediment size. *Marine Geology*, *417*, 106012. https://doi.org/10.1016/j.margeo.2019.106012

Vos, K., Harley, M. D., Splinter, K. D., Walker, A., & Turner, I. L. (2020). Beach Slopes From Satellite-Derived Shorelines. *Geophysical Research Letters*, *47*(14), e2020GL088365. https://doi.org/10.1029/2020GL088365
_eprint: https://agupubs.onlinelibrary.wiley.com/doi/pdf/10.1029/2020GL088365

---

## Author Response (AR1)

**ESSD-2021-388 – RC1 comment by Floris Calkoen**

We would like to thank the referee for his excellent review, very concise summary and insightful comments on the methodology, limitations, and dataset structure. We appreciate the fact that the referee took the time to download the Github repository and directly test the methodology.

**1. General comments**

**1.1. Revisit period and peak tidal frequency**

The irregular sampling interval is a very important point, which needs clarification in the manuscript for the benefit of the readers. Three satellites from the Landsat constellation are used in CoastSat to map shorelines: Landsat 5 (1984-2013), Landsat 7 (1999-current), Landsat 8 (2013-current). Theoretically, each satellite has a revisit period of 16 days for every point on Earth (below 80 degrees latitude), so when two satellites are in orbit simultaneously (1999-2019), we should have an 8-day revisit period. As the referee stated, this 8-day sampling interval is the key to this algorithm, as it allows to capture, with a certain degree of aliasing, the fortnightly tidal cycles. Now in practice, there are several factors that, to our knowledge, alter the revisit period of Landsat images (only talking about the images here, not the shorelines):

i) *Swath overlap*: the Landsat satellites are in a near-polar orbit, where the orbit paths come closer together near the poles. This means that the overlap between images from adjacent swaths increases with latitude, from 7% at the Equator to a maximum of 85% at high latitudes (Department of the Interior U.S. Geological Survey, 2016). Thus, regions located at the edge of the swath will be imaged more frequently.
The Landsat scene boundaries (available at https://www.usgs.gov/media/files/landsat-wrs-2-scene-boundaries-kml-file), displayed in Figure 1, show that Cable Beach is located in a region of high overlap (both row and path) hence the 0-days and 7-days revisits in addition to the default 8-days revisit period (as shown by the referee's figure) .

ii) *Availability in Tier 1:* the Landsat images are sorted into Tier 1 and Tier 2 based on whether they meet or not formal geometric and radiometric quality criteria (USGS, 2017). The data provider specifies that only Tier 1 images, which pass the geometric and radiometric quality control, are suitable for time-series analysis.

iii) *Acquisition strategy:* as the referee rightly pointed out, beyond the Australian scale, the acquisition strategy of the Landsat program was not even across the globe (Wulder et al., 2016), especially for the earlier missions (Landsat 5 and Landsat 7) which required ground receiving stations in line of sight. This results in substantial data gaps in certain regions.

[Figure]

*Figure 1. Landsat Scene Boundaries showing the regions of overlap, and the location of Cable Beach in Western Australia.*

**Response to a)**: The factors mentioned above are the ones that to our knowledge affect the image availability. In addition, there are the issues with the shoreline detection, like clouds, misclassifications, outliers, which also contribute to the unequal temporal distribution of the shoreline time-series.

We have added a sentence briefly describing the reasons for the irregular sampling of the shoreline time-series at **Lines 91-95**. The acquisition strategy and applicability of this method to other geographical areas is addressed in the Discussion.

The 20-year time period (1999-2020) is selected to ensure that two Landsat satellites are in orbit simultaneously, which theoretically results in a combined revisit period of 8 days (16 days for each satellite). However, in practice, time-series of shoreline change are irregularly sampled due to factors such as cloud cover, gaps in Landsat 7 data (scan line corrector error) and discarded images due to poor geometric or radiometric quality. Any outliers in the shoreline time-series (e.g., due to false detections) are also removed using a despiking algorithm.

**Response to b)**: The Jupyter Notebook shows an example of how to estimate the beach slope at an individual site. The user can manually select the main sampling period based on the timestep distribution (Figure 2a). If we choose 7 days (Nyquist limit is 14 days), the power spectrum of the tide levels shows a peak at 14.8 days but also a smaller peak at 17.5 days, which results from the aliasing of the 14.8 days cycle with an 8-day sampling period (Figure 2b, also see demonstration of this aliasing in Vos et al. (2020) Supporting Information S3). The algorithm always chooses the frequency with the highest peak on the tidal spectrum to estimate the slopes (with a hard-coded limit at 1 month to only consider high frequencies). If we choose 8 days, the Nyquist limit is 16 days and only the aliased 17.5 days peak is present on the power spectrum (Figure 2c). Note that as we are using a least-squares spectral method (Lomb-Scargle transform), we can design the frequency grid and set its limits and spacing (see discussion in Specific Comments 2).

Now in this example, we get the same slope estimates, whether we use a 7- or 8-days image acquisition interval, we invite the referee to try this on the notebook by changing `settings_slope['n_days']=8`. As we didn't observe any differences in this case (only site amongst our validation sites to be in an area of overlap), we have processed the entire dataset with an image acquisition interval of 8 days, which is the theoretical sampling period for two Landsat satellites in orbit simultaneously and should be the most common interval between images.

[Figure]

Figure 2. *a)* Distribution of the interval between consecutive images. *b)* Power spectrum density of the tide levels associated with the images using a main sampling period of 7 days (Nyquist limit of 14 days). *c)* same with a main sampling period of 8 days (Nyquist limit of 16 days).

We have addressed this procedure in more detail in the manuscript and have expanded at **lines 98-105**:

Since the Lomb-Scargle is a least-squares spectral method, the limits and spacing of the frequency grid need to be first defined. Here, a main sampling period of 8 days (i.e., the theoretical Landsat revisit period) is used, resulting in a maximum frequency (Nyquist limit) of 16 days, and the spacing n_0 is set to 50 samples per peak to ensure that the grid sufficiently samples each peak (VanderPlas, 2018, Section 7.1). In the resulting PSD, the frequency with the highest peak is isolated, which corresponds to the frequency where the tidal signal (e.g., the spring-neap tidal cycle) is strongest in the sub-sampled time-series and is referred to as the 'peak tidal frequency'. Note that the spring-neap cycle has a period of 14.8 days but is subject to aliasing when sampled at an 8-day interval resulting in a peak at 17.5 days (refer Supporting Information S3 in Vos et al., 2020).

**Response to c)**: the 0-days interval are present in areas of *path overlap* (overlap between images along the same path), while the 7-days interval are areas of *row overlap* (overlap between two adjacent rows).

**Response to d)**: the outlier removal step is important as outliers can introduce errors in the computation of the spectra. Note that the `SDS_slope.reject_outliers()` function that is commented in the Jupyter notebook is setup for Narrabeen-Collaroy (a microtidal environment) and is not valid for Cable Beach (macrotidal). We created an *ad hoc* despiking algorithm which looks to remove points in the shoreline time-series that are physically impossible. For example, a sandy beach can be expected to erode of tens or even hundreds of metres in one timestep as a result of storm erosion (or accrete as a result of a nourishment or sand deposition), however it will take time to

recover that beach width, so if we have a spike in the time-series where the shoreline eroded X metres and accreted X metres in the next timestep, this has to be a false detection. The parameter regulating this distance X is the `max_cross_change` despiking parameter and is adjusted as a function of tidal range to allow for the tidal excursion for consecutive images acquired at different tidal stages. Hence the parameter is set to 40 m for beaches with a spring tidal range below 3 m and then follows a linear relationship as shown in Figure 3 (this relationship assumes tidal excursion with a slope of 0.02 to be conservative). According to this relationship `max_cross_change` should be 300 m for Cable Beach, which leads to 0 outliers being removed in the Jupyter notebook.

[Figure]

*Figure 3. Relationship between the value of the despiking parameter and the Mean Spring Tidal Range.*

We have added a short sentence to describe this step at **lines 94-95**:

Any outliers in the shoreline time-series (e.g., due to false detections) are also removed using a despiking algorithm.

And guided the readers to the Jupyter notebook examples for full details on the applied methodology, **lines 109-110**:

A full example of this procedure is presented for both a microtidal site (Narrabeen-Collaroy) and a macrotidal site (Cable Beach) in the form of Jupyter notebooks at https://github.com/kvos/CoastSat.slope (Vos, 2021b).

**1.2. Limitations**

**Response to a) and b)**:

We thank the reviewer for this very good point, which we had not considered before. We agree with the referee that, based on the reasoning and references mentioned, Australia is a very favourable location for optical remote sensing with Landsat images.

We have added a paragraph in the discussion section following the referee's reasoning, stressing the limitations associated with applying this method in other geographical areas due to image availability in the Landsat archive and cloud frequency.

**Lines 257-262**:

While the focus of this dataset is on the Australian coastline, the generic nature of the method and global extent of Landsat imagery means that such a dataset could also be theoretically reproduced elsewhere. It is important to note however that the density of Landsat coverage is not consistent globally (Wulder et al., 2016) and that Australia (along with North America and Eastern China) has

some of the highest coverage in terms of image density. Additionally, the Australian continent is characterised by a relatively low mean cloud frequency (Wilson and Jetz, 2016), which means that a lower proportion of optical images are hindered by clouds. Other areas with a sparser Landsat coverage and/or higher cloud frequency, may not have the same temporal depth of shoreline observations, which could hinder the applicability of this method in some regions.

**1.3.  Data structure**

**Response to a)**: we have renamed all the fields using underscores and lower cases consistently.

**Response to b)**: we have adjusted Tables 1 and 2 to match the identifiers in the dataset.

**Response to c)**: it is a good suggestion to integrate the dataset into a database format. For the moment we think that a geospatial layer is suitable for the coastal sciences audience, which is more likely to import it into QGIS, but will consider the STAC framework for future developments.

**2.  Specific comments**

*2.1.* The initial transect dataset was downloaded in 2018, and since it was quality-controlled and trimmed by manual digitalisation, we could not reproduce it automatically for the 2021 version of the OSM database.

*2.2.* This is a good question, while we are not experts in signal processing, our understanding is that a Discrete Fourier Transform (DFT) requires a convolution over different windows which will be distorted by the irregular sampling frequency. VanderPlas, (2018) discusses this issue in *Section 4. Nonuniform sampling*. While this issue can be mitigated by densely sampling the signal, albeit non-uniformly, this is not possible in our case as we must work with relatively large sampling intervals (days to weeks). The Lomb-Scargle transform is a least-squares spectral method that was specifically developed by astronomers to characterise periodicity in unevenly sampled time-series from optical sensors. It also has the advantage that we can define the frequency grid (limits and spacing), rather than it be dictated by the number of samples as in DFT/FFT. This is useful in our application as we can set a fine grid spacing around the frequencies of interest (i.e., fortnightly cycle) to fully capture those peaks and make sure they do not fall between grid points.

**3. Technical corrections**

**3.1.  Textual**

We thank the referee for the suggestions, we have been incorporated them into the revised manuscript.

The last section of the manuscript (previously *Discussion and Outlook*) was split into a Discussion section (3 paragraphs) and a concise one-paragraph Conclusion.

Discussion (**lines 238-262**):

[revised manuscript text omitted]

**ESSD-2021-388 – RC2 comment by Giovanni Scicchitano**

We would like to thank the referee for their positive comments and suggestion for improving the discussion/conclusion parts.

We have restructured the last section (Discussion and Outlook) and split it into a 3-paragraph Discussion and one-paragraph Conclusion.

Discussion (**lines 238-262**):

[revised manuscript text omitted]

**ESSD-2021-388 – CC1 comment by Robbi Bishop-Taylor**

We would like to thank Dr. Bishop-Taylor for the careful and thorough reading of this manuscript and for the thoughtful comments and constructive suggestions.

We did indeed followed the concept of tidal aliasing by sun-synchronous sensors that was introduced in the previous work by Bishop-Taylor et al. (2019) and were happy to find very similar patterns (with as low as 70% of tidal coverage, equivalent to the 30% bias reported in their analysis).

It is a very good suggestion to complement the *% MSTR observed* with the actual range of tidal elevations that are covered by the Landsat imagery. We have included this as additional metadata in the dataset (*min_tide_obs* and *max_tide_obs*) and agree that this will provide useful information to end-users.

There a several reasons why we believe that the beach-face slope estimated with our method is a proxy for the slope from MSL to MHWS, rather than from MLWS to MHWS, we have provided our arguments below and have clarified it in the manuscript to the benefit of the readers.

Initially, when we performed the validation against *in situ* surveys (across 8 sites with long-term survey data), we found that the slope estimates obtained with our algorithm matched best with the MSL to MHWS slope, while they tended to overestimate the MLWS to MHWS. While this observation is only data-driven, there is a physical rationale to support it.

Firstly, the beach-face is usually not linear (as assumed in our method) but rather concave and therefore can be described by two slopes, a steeper upper beach-face slope (MSL to MHWS) and a flatter lower beach-face slope (MLSW to MSL). The upper part tends to be more stable over time, while the lower beach-face slope is very dynamic as surfzone bars detach and attach to the shoreline, making it suddenly steeper or flatter (Wright and Short, 1984). For example, a transition from an intermediate state (e.g., transverse bar and rip) to a low tide terrace will drastically change the lower beach-face slope, but only affect the upper beach-face slope to a lesser extent.

To illustrate this point, the figure below shows the measured 40-year time-series for the lower (MLWS-MSL) and upper (MSL-MHWS) beach-face slopes at Narrabeen-Collaroy (Profile 1). You can see how the MLWS-MSL slope (orange) is flatter and more noisy (i.e., wider distribution) than the MSL-MHWS slope.

[Figure]

Therefore, we believe that this may explain why the resulting long-term slope estimate from our algorithm matches better with the beach-face slope (MSL to MHWS) which is more stable over time and a better representation of the 'typical' beach-face slope for that site. Unfortunately, the resolution of the publicly available images is not fine enough to estimate both the upper and lower beach-face slope at our study sites, but this could become possible with higher resolution commercial sensors

(Planet Labs, Maxar etc). We believe that using MSL-MHWS as proxy for the beach-face slope is somewhat analogous to using the MHWS-contour to monitor shoreline changes instead of the MSL-contour which is more noisy (Castelle et al., 2014; Splinter et al., 2014).

A second point to consider, is that the algorithm uses astronomical tides to calculate the water levels at which the Landsat images were acquired. However, the instantaneous water levels are also affected by wave setup (along wave-dominated beaches) and/or wind setup (along tide-dominated beaches), which tend to add a positive bias to the astronomical predictions. While presently these effects cannot be incorporated at the continental-scale due to the lack of nearshore hydrodynamic data, they may also contribute to the fact that our slope estimates match better with the upper beach-face slope rather than the entire intertidal zone. A positive vertical bias is also used to adjust the CoastSnap-derived shorelines (Harley et al., 2019) and Argus-derived shorelines (Harley et al., 2011).

**Lines 115-119**:

The satellite-derived beach-face slope estimates were found to match best with the slope between MSL to MHWS (R2 = 0.93, bias = 0.0), while they tended to overestimate the full intertidal (MLWS to MHWS) slope. This can be explained by the fact that the upper intertidal slope (MSL to MHWS) is generally more stable over time, while the lower intertidal slope (MLWS to MSL) is more variable as intertidal bars attach/detach to the shoreline (Wright and Short, 1984). Wave runup and setup effects that are not included in the global tide model also tend to skew the shoreline detection towards the upper part of the intertidal profile (e.g., Harley et al., 2019).

We also thank Dr. Bishop-Taylor for the technical corrections, we have documented the missing fields and reprojected the geospatial layers to WGS84 as suggested.